# Specific Eph receptor-cytoplasmic effector signaling mediated by SAM–SAM domain interactions

Yue Wang[1], Yuan Shang[2†], Jianchao Li[2], Weidi Chen[1], Gang Li[1], Jun Wan[1,2], Wei Liu[1]*, Mingjie Zhang[1,2]*

[1]Shenzhen Key Laboratory for Neuronal Structural Biology, Biomedical Research Institute, Shenzhen Peking University-The Hong Kong University of Science and Technology Medical Center, Shenzhen, China; [2]Division of Life Science, State Key Laboratory of Molecular Neuroscience, Hong Kong University of Science and Technology, Kowloon, China

*For correspondence:
liuwei@sphmc.org (WL);
mzhang@ust.hk (MZ)

Present address: [†]Center for Biomedical Informatics and Biostatistics & Center for Innovation in Brain Science, The University of Arizona, Tucson, Arizona

**Abstract** The Eph receptor tyrosine kinase (RTK) family is the largest subfamily of RTKs playing critical roles in many developmental processes such as tissue patterning, neurogenesis and neuronal circuit formation, angiogenesis, etc. How the 14 Eph proteins, via their highly similar cytoplasmic domains, can transmit diverse and sometimes opposite cellular signals upon engaging ephrins is a major unresolved question. Here, we systematically investigated the bindings of each SAM domain of Eph receptors to the SAM domains from SHIP2 and Odin, and uncover a highly specific SAM–SAM interaction-mediated cytoplasmic Eph-effector binding pattern. Comparative X-ray crystallographic studies of several SAM–SAM heterodimer complexes, together with biochemical and cell biology experiments, not only revealed the exquisite specificity code governing Eph/effector interactions but also allowed us to identify SAMD5 as a new Eph binding partner. Finally, these Eph/effector SAM heterodimer structures can explain many Eph SAM mutations identified in patients suffering from cancers and other diseases.
DOI: https://doi.org/10.7554/eLife.35677.001

## Introduction

The Eph (erythropoietin-producing hepatocyte) transmembrane receptor tyrosine kinase superfamily, with its first member identified 30 years ago (*Hirai et al., 1987*), contains 14 members in mammals and is the largest among all receptor tyrosine kinase families (*Lemmon and Schlessinger, 2010*; *Manning et al., 2002*; *Murai and Pasquale, 2003*). Chiefly based on their engaging ephrin ligands, Eph receptors are classified into the EphA and EphB subfamilies, each with nine and five members in mammals, respectively (*Eph Nomenclature Committee , 1997*; *Gale et al., 1996*; *Murai and Pasquale, 2003*). Owing to broad expressions in essentially all tissues and at every life stage, Ephrin-Eph signaling regulates many cellular processes both during development and in developed animals such as stem cell maintenance and differentiations, tissue morphogenesis, and tissue–tissue boundary formation (*Batlle and Wilkinson, 2012*; *Genander and Frisén, 2010*; *Jülich et al., 2009*; *McMillen and Holley, 2015*; *Munarini et al., 2002*; *Park et al., 2011*; *Poliakov et al., 2004*). Not surprisingly, mutations of ephrins and Ephs are known to cause many forms of diseases including cancers and brain disorders (*Boyd et al., 2014*; *Chen et al., 2008*; *Hahn et al., 2012*; *Kania and Klein, 2016*; *Merlos-Suárez and Batlle, 2008*; *Pasquale, 2008*; *Zhuang et al., 2012*).

Ephrin ligand binding-mediated inter-cellular signaling is the classic mode of Eph receptor signaling (also known as ephrin-Eph 'forward' signaling), which is responsible for the majority of cellular functions characterized for the ephrin-Eph signaling (*Pitulescu and Adams, 2010*; *Taylor et al.,*

**eLife digest** As an animal's body develops, its cells need to find their way to the right place to form its tissues and organs. On top of this, nerve cells need to set up connections as they grow. A family of receptors called Eph receptors help to make this happen. They sit across cell membranes, waiting for signals from molecules called ephrins. Once activated, these receptors interact with other proteins inside the cell.

There are 14 different Eph receptors, but the parts inside the cell are similar, with three domains arranged in a set order. Next to the membrane, there is a tyrosine kinase domain, an enzyme that can add a phosphate group to a protein. Then, there is a SAM domain, which interacts with other proteins. Finally, there is a PDZ domain binding motif, which anchors the receptor to the cell's internal skeleton.

The similarity between the internal portions of the Eph receptors suggests that they should work in the same way. But, different receptors on the same cell, responding to the same external signal, can have opposite effects. Here, Wang et al. tested each of the 14 SAM domains to find out how this happens.

SAM domains on Eph receptors interact with SAM domains on other proteins, including SHIP2 and Odin. Analysis of the interactions revealed specific patterns for each receptor. Even though SAM domains are similar in shape, their exact amino acids – the basic building blocks of proteins – differ at particular positions. This changes the way they interact, allowing them to bind to different partners.

Wang et al. then used a technique called X-ray crystallography to reveal the three-dimensional structures of SHIP2 bound to EphA2 and Odin bound to EphA6, to see how the proteins interact in fine detail. It turns out that a piece of each Eph receptor called the "end helix" binds to a "mid-loop" structure in SHIP2 or Odin. Crucial amino acids in each ensure that these interactions are specific. Changing these critical positions prevented the proteins coming together or allowed them to bind to a completely different partner.

The structures revealed the importance of negatively charged amino acids within the mid-loop of the Eph binding partners. Using this information, Wang et al. predicted and confirmed a brand-new interaction between EphA5 and one of the 127 SAM-containing proteins found in mice, a protein called SAMD5.

Understanding the impact of protein structure on Eph receptors could aid research into human disease. Lastly, an analysis of a database containing genetic changes found in cancer patients revealed that many of the mutations occur inside SAM domains. Pinpointing the positions that affect Eph receptor binding could point the way to future treatments.

DOI: https://doi.org/10.7554/eLife.35677.002

*2017*; *Yokoyama et al., 2001*). Presumably, the versatile forward ephrin-Eph signals are transmitted by the cytoplasmic portion of Eph receptors. However, the cytoplasmic portion of all 14 Eph receptors are highly similar, each containing a membrane-juxtaposing kinase domain, a protein-binding SAM domain immediately followed by a short carboxyl tail PDZ domain-binding motif (PBM) (*Figure 1A*). The cytoplasmic portions of Eph receptors are often presumed to function similarly. However, multiple Eph receptors are typically co-expressed in one tissue. Paradoxically, it has been observed that two different Eph receptors on the same cell type can respond to a single ephrin ligand but elicit opposite cellular responses. For example, ephrin-A5 binds to EphA2 and EphA4 with similar affinity but induces cell adhesion or cell collapse, respectively (*Cooper et al., 2008*; *Zhou et al., 2007*), although different multimerization modes of EphA2 and EphA4 extracellular domains induced by ephrin-A5 binding can also contribute to the opposite cell spreading phenotype (*Seiradake et al., 2013*). Therefore, the cytoplasmic domains of Eph receptors must be able to engage different intracellular effectors in response to ephrin ligands. How specific Eph receptor cytoplasmic domain-mediated signaling might occur has been a major unresolved question in the ephrin-Eph signaling.

We reasoned that the SAM domain of each Eph receptor is likely to play a role in specifying their cytoplasmic effector engagements for the following two reasons. First, SAM domain is a well-known

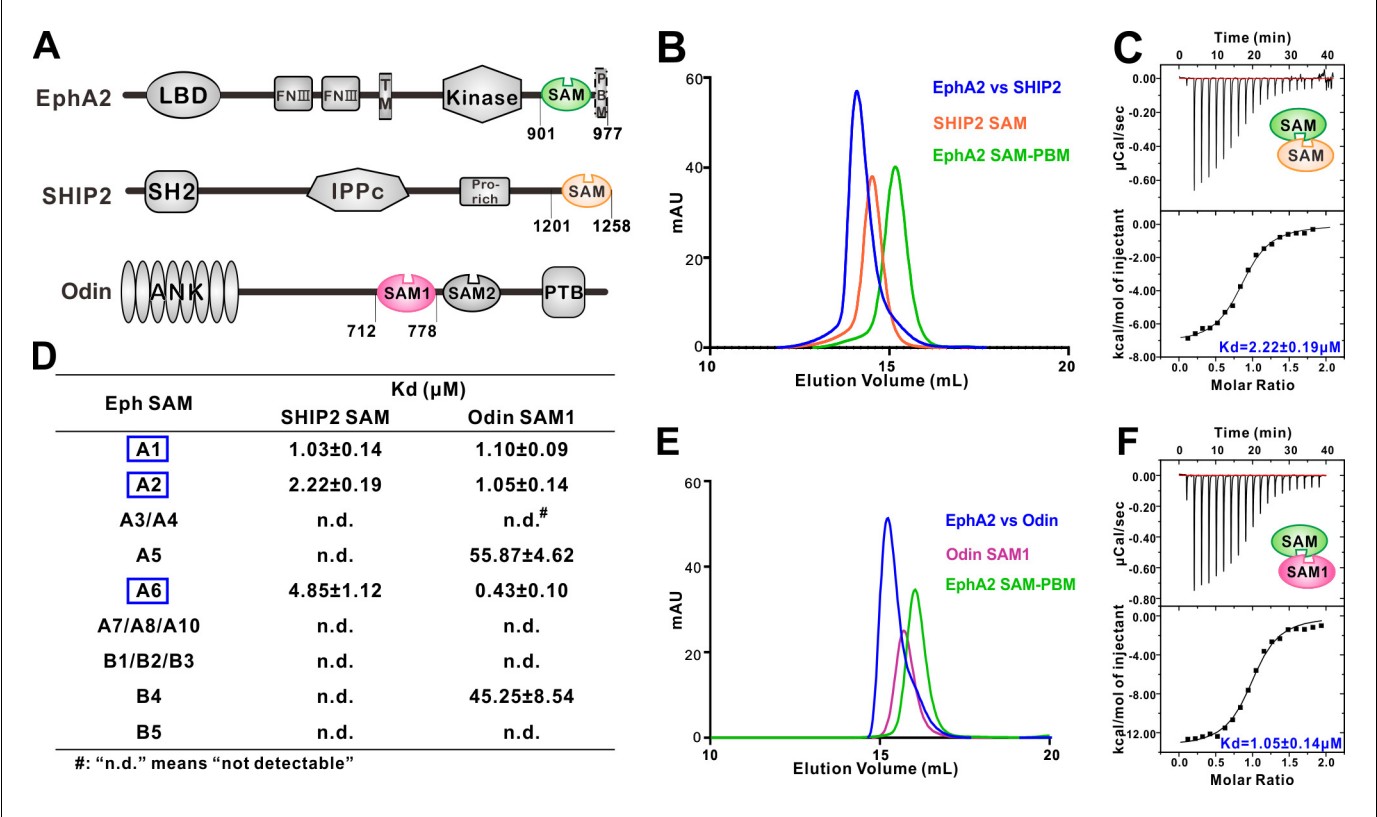

**Figure 1.** Several EphA SAM domains specifically bind to SHIP2 SAM or Odin SAM. (**A**) Schematic diagrams showing the domain organizations of EphA2, SHIP2 and Odin. (**B–C**) Analytical gel-filtration chromatography (**B**) and ITC-based measurements (**C**) showing the interaction between EphA2 SAM and SHIP2 SAM. (**D**) Summary of the dissociation constants between Eph SAMs and SHIP2/Odin SAMs. All binding affinities were derived from ITC-based assays. (**E–F**) Analytical gel-filtration chromatography (**E**) and ITC-based measurements (**F**) of the binding of EphA2 SAM to Odin SAM.
DOI: https://doi.org/10.7554/eLife.35677.003

protein–protein interaction module (*Qiao and Bowie, 2005*), and the SAM domain of EphA2 is known to bind to SAM domain from SHIP2 (SH2 domain-containing Inositol 5'-Phosphatase 2, aka INPPL1 for INositol Polyphosphate Phosphatase-Like protein 1) and Odin (aka Anks1a) (*Kim et al., 2010*; *Lee et al., 2012*; *Leone et al., 2009*; *Mercurio et al., 2012*; *Zhuang et al., 2007*), though the binding properties of the SAM domains from other Eph receptors are largely unknown. Second, although the PBM sequences of Eph receptors are somewhat different, the short PBM-mediated target bindings are rather promiscuous (*Ye and Zhang, 2013*) and thus unlikely to be fully responsible for the very diverse Eph intracellular signaling events. In this study, we systematically characterized and compared the bindings of the SAM domain from every Eph receptor to the SAM domains from SHIP2 and Odin. This characterization revealed a highly specific Eph SAM and effector SAM-binding pattern. We then elucidated the mechanistic basis governing such specific Eph SAM and effector SAM binding by solving several pairs of the SAM-SAM heterodimer complexes structures. Such comparative structural analysis, together with biochemical, bioinformatics and cell biology studies, revealed an exquisitely specific effector binding code mediated by the Eph SAM domains, which helps to answer the major question on the ephrin-Eph forward signaling specificity. Additionally, our study also provides mechanistic explanations to numerous disease-causing mutations identified in the SAM domains of Eph receptors, and allows us to discover SAMD5 as a new intracellular effector of Eph receptors.

## Results

### Interactions between Eph SAM domains and SHIP2 SAM or Odin SAM1

SHIP2 is a mammalian inositol polyphosphate 5-phosphatases and is the only member in the family that contains a C-terminal SAM domain (*Figure 1A*). It has been reported that SHIP2 was a binding partner of EphA2 through SAM–SAM interaction (*Lee et al., 2012*; *Leone et al., 2009*). We first confirmed this interaction. Both Eph SAM and SHIP2 SAM alone behaved as homogeneous monomers in solution as indicated by the analytical gel filtration analysis (*Figure 1B*). While the 1:1 mixture of EphA2 SAM and SHIP2 SAM was eluted at a smaller volume than the individual proteins, suggesting the formation of a hetero SAM-SAM complex (*Figure 1B*). ITC (Isothermal Titration Calorimetry) experiment revealed that EphA2 SAM bound to SHIP2 SAM with a dissociation constant (Kd) of ~2.22 µM at a 1:1 stoichiometry (*Figure 1C*). We then measured the binding affinities of SHIP2 SAM with the SAM domain from other members of Eph receptors by ITC. We found that only the SAM domains of EphA1/EphA2/EphA6 specifically bound to SHIP2 SAM, and the rest of Eph SAM domains displayed no detectable binding to SHIP2 SAM (*Figure 1D*).

Odin was reported to be another binding partner of EphA2 (*Kim et al., 2010*; *Mercurio et al., 2012*), and the interaction had been implicated in affecting the stability of EphA2 by modulating its ubiquitination process (*Kim et al., 2010*). We also verified the interactions between Odin and Eph receptors SAM domains (*Figure 1E and F*). The results showed that, similar to SHIP2 SAM, Odin SAM1 only selectively bound to the SAM domains from EphA1/EphA2/EphA6 (*Figure 1D*). The quantitative and systematic binding results shown in *Figure 1D* indicated that the interactions between Eph SAMs and their downstream effectors were highly specific.

### Overall structures of the EphA2/SHIP2 and EphA6/Odin SAM-SAM complexes

To elucidate the molecular mechanism governing the specificity of the bindings of SHIP2 SAM and Odin SAM1 to Eph receptors, we determined the crystal structure of the SHIP2 SAM-EphA2 SAM complex and the Odin SAM1-EphA6 SAM complex at the 1.5 Å and 1.3 Å resolutions, respectively (*Table 1*). The crystals diffracting at very high resolutions of both complexes were facilitated by fusing the Eph SAM domain to the C-terminal tail of the SHIP2 SAM or Odin SAM1 with a flexible linker (14 residues 'SSGENLYFQSGSSG' for the SHIP2 SAM-EphA2 SAM complex; 17 residues 'PSGSSGE NLYFQSGSSG' for the Odin SAM1-EphA6 SAM complex). The covalent linkage did not appear to affect the overall structure of the complexes, as the linkers in both complexes are sufficiently long and flexible (*Figure 2—figure supplement 1*).

The two complex structures adopt an essentially identical End-Helix/Mid-Loop binding mode, in which positively charged residues from the N-terminal end of α5 (End-Helix) of EphA2/EphA6 bind to negatively charged residues from the loop connecting α2-α4 (Mid-Loop) from SHIP2/Odin, forming the well-known tail-to-head SAM domain heterodimer (*Figure 2A–E*) (*Qiao and Bowie, 2005*). The structures determined in this study were also consistent with an earlier NMR-derived EphA2 SAM/SHIP2 SAM heterodimer (*Lee et al., 2012*; *Leone et al., 2009*). However, it should be noticed that some of the critical features mediating the specific SAM–SAM interaction revealed in our study were not revealed in the NMR structures (*Figure 2* and *Figure 2—figure supplement 2*), likely due to insufficient distance restraints of the NMR experiments. To avoid redundancy, we will not describe the detailed binding interactions of the two complexes here, except that we further validated some of such charge–charge interactions using site substitution approach (*Figure 2H*). The structures of the complexes also revealed that the surfaces of the EphA2/EphA6 SAM Mid-Loop do not complement with the surfaces of the End-Helix of their own or with those of the SHIP2/Odin SAM domains (*Figure 2—figure supplement 3*), explaining that these four SAM domains neither form homo-oligomers nor polymerize into hetero-oligomers (*Knight et al., 2011*; *Qiao and Bowie, 2005*; *Stapleton et al., 1999*; *Thanos et al., 1999*). A noticeable feature in both complexes is that the backbone methylene of a Gly residue at the beginning of α5 from Eph SAM (Gly954$^{A2}$/Gly1104$^{A6}$) is in close contact with an aromatic residue from SHIP2/Odin SAM (Trp1221$^{SHIP2}$/Phe738$^{Odin}$) (*Figure 2B and D* and *Figure 2—figure supplement 4*). As such, replacing Gly at the beginning of α5 of Eph SAM with any other amino acid residues will introduce steric hindrance in preventing their binding to SHIP2/Odin SAM. It is further noted that Gly is highly preferred at the beginning of α5

**Table 1.** Statistics of X-ray crystallographic data collection and model refinement

**Data collection**

| Dataset | EphA2/SHIP2 | EphA6/Odin | EphA5/SAMD5 |
|---|---|---|---|
| Space group | $C2$ | $P2_12_12_1$ | $C2$ |
| Unit cell (a, b, c, Å) | 138.283, 43.344, 46.377 | 38.985, 85.042, 98.383 | 98.195, 29.651, 54.170 |
| Unit cell (α, β, γ, °) | 90, 95.354, 90 | 90, 90, 90 | 90, 109.952, 90 |
| Wavelength (Å) | 0.97915 | 0.97915 | 0.97961 |
| Resolution range (Å) | 50.00–1.50 (1.53–1.50) | 50.00–1.30 (1.32–1.30) | 50.00–1.90 (1.93–1.90) |
| No. of unique reflections | 43485 (2131) | 79398 (3792) | 11829 (575) |
| Redundancy | 3.6 (3.5) | 5.9 (5.7) | 3.2 (3.2) |
| I/σ | 37.9 (2.4) | 39.2 (2.1) | 25.7 (1.9) |
| Completeness (%) | 99.0 (98.1) | 97.8 (95.1) | 99.1 (100.0) |
| $R_{merge}$ (%)* | 4.7 (62.4) | 5.4 (90.3) | 5.3 (77.1) |
| **Structure refinement** | | | |
| Resolution (Å) | 50.0–1.50 (1.55–1.50) | 50.0–1.30 (1.35–1.30) | 50.0–1.90 (1.96–1.90) |
| $R_{cryst}$[†]/$R_{free}$[‡] | 0.1434 (0.2686)/ 0.1911 (0.3129) | 0.1695 (0.2565)/ 0.1987 (0.2671) | 0.1938 (0.3256)/ 0.2340 (0.3282) |
| Rmsd bonds (Å)/angles (°) | 0.009/1.22 | 0.010/1.52 | 0.015/1.67 |
| Average B factor (Å²)[§] | 30.8 | 22.8 | 30.1 |
| No. of protein atoms | 2046 | 2745 | 1058 |
| No. of other atoms | 273 | 443 | 96 |
| No. of reflections | | | |
| Working set | 41244 (4003) | 77272 (7372) | 11252 (1065) |
| Test set | 2186 (227) | 1947 (190) | 565 (57) |
| Ramachandran plot (%)[§] | | | |
| Favored regions | 99.2 | 99.1 | 98.5 |
| Allowed regions | 0.8 | 0.9 | 1.5 |
| Outliers | 0 | 0 | 0 |

Numbers in parentheses represent the value for the highest resolution shell.

* $R_{merge} = \Sigma|I_i - I_m|/\Sigma I_i$, where $I_i$ is the intensity of the measured reflection and $I_m$ is the mean intensity of all symmetry-related reflections.

† $R_{cryst} = \Sigma||F_{obs}| - |F_{calc}||/\Sigma|F_{obs}|$, where $F_{obs}$ and $F_{calc}$ are observed and calculated structure factors.

‡ $R_{free} = \Sigma_T||F_{obs}| - |F_{calc}||/\Sigma_T|F_{obs}|$, where T is a test data set randomly chosen and set aside prior to refinement. For EphA6-Odin, 2.5% of total reflections were enough for the test set during refinement. For EphA2-SHIP2 and EphA5-SAMD5, 5% of total reflections were chosen as test sets.

§ B factors and Ramachandran plot statistics are calculated using MOLPROBITY (*Chen et al., 2010*).

DOI: https://doi.org/10.7554/eLife.35677.004

among Eph SAM domains (12 out of the 14 members are Gly; see *Figure 2—figure supplement 5A*).

The very high-resolution crystal structures of the two complexes also allowed us to identify a unique interaction feature that is critical for the exquisite specific interaction between the EphA2/A6 SAM domain and SHIP2/Odin SAM domains. Taking the EphA2/SHIP2 complex as the example, a special cation-π interaction between the Arg958[A2] and Phe1226[SHIP2] was uncovered by the high-resolution crystal structure (*Figure 2F*). The guanidinium group of Arg958[A2] forms hydrogen bond network with Asp1222[SHIP2] and His955[A2] on one side and with the backbone carbonyl oxygen of Ile917[A2]. As such the guanidinium plane of Arg958[A2] (and its delocalized π-system) is in the same plane with the π-system of the planer peptide bond between Ile917[A2] and Lys918[A2], forming energetically favorable π-π stacking with the benzene ring of Phe1226[SHIP2] (*Figure 2F*) (*Ma and Dougherty, 1997*). The exactly same interaction pattern occurs for the EphA6/Odin complex (*Figure 2G*).

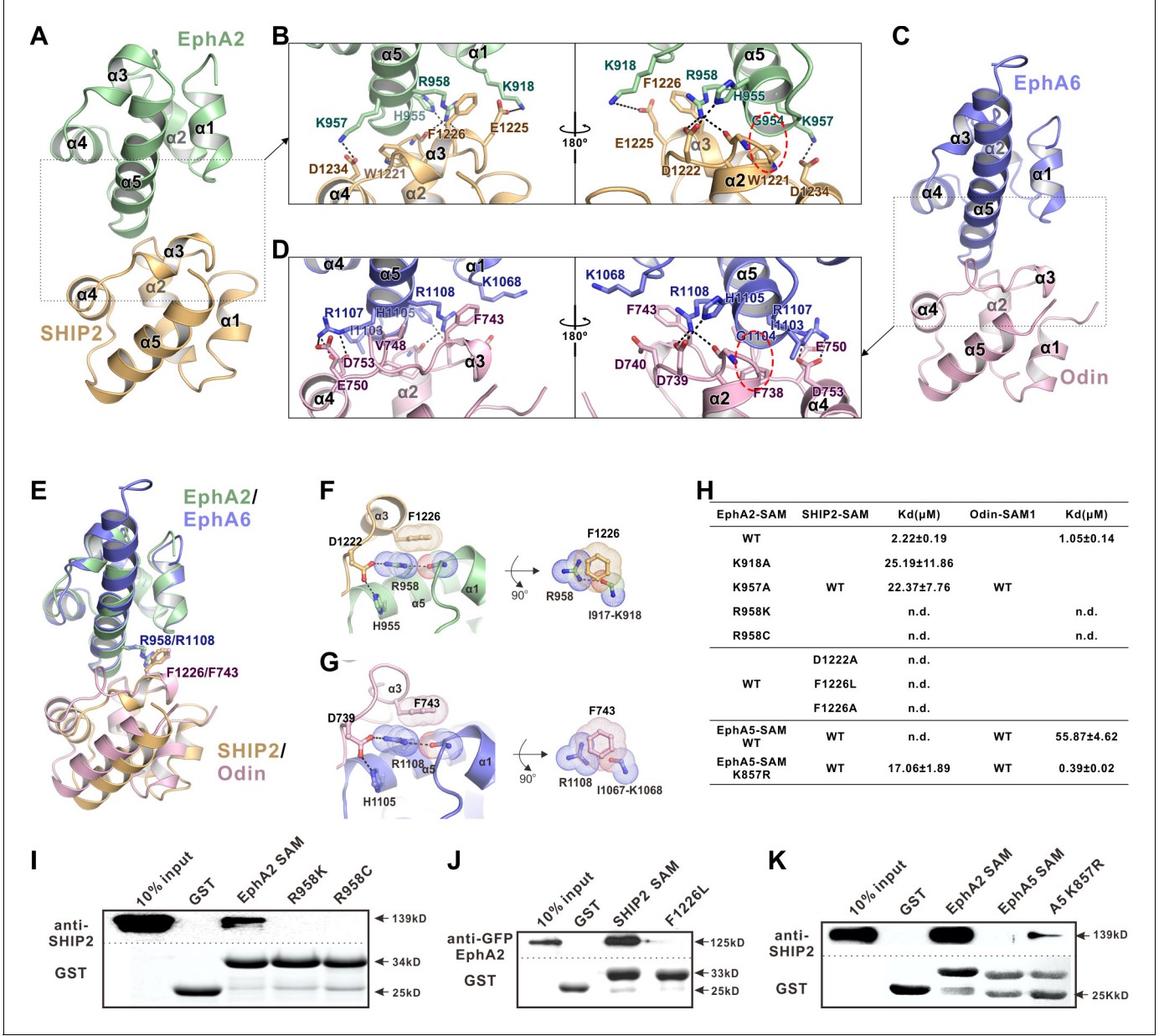

**Figure 2.** The detailed interactions governing the formation of the EphA2 SAM-SHIP2 SAM and the EphA6 SAM-Odin SAM1 complexes. (**A and C**) Ribbon representations of the EphA2 SAM-SHIP2 SAM (**A**) and the EphA6 SAM-Odin SAM1 (**C**) complex structures. (**B and D**) Details of the interfaces in the EphA2 SAM-SHIP2 SAM (**B**) and EphA6 SAM-Odin SAM1 (**D**) complexes. Key residues are shown with the stick model. Salt bridges and hydrogen bonds are indicated with dashed lines. The Gly-aromatic residue pairs (Gly954[A2]/Gly1104[A6] and Trp1221[SHIP2]/Phe738[Odin]) are indicated with red dashed circles. (**E**) Superposition of two the EphA2 SAM-SHIP2 SAM and the EphA6 SAM-Odin SAM1 complex structures with the critical R958/F1226 and R1108/F743 pairs highlighted with the stick model. (**F–G**) Details of the planar cation-π interaction between EphA2 R958 and SHIP2 F1226 (**F**), and EphA6 R1108 and Odin F743 (**G**). Note the planar alignment of the π electrons of the Arg sidechain and the neighboring peptide backbones in both structures. (**H**) Summary of the ITC-derived results showing that the mutations of the critical residues in the SAM domain interface can dramatically affect the bindings of EphA SAMs to the SAM domains from SHIP2 or Odin. (**I**) The Bindings of EphA2 SAM or its various mutants to the full-length SHIP2 by the GST pull-down assay. (**J**) The bindings of SHIP2 SAM or its F1226L-mutant to the full-length WT GFP-EphA2 analyzed by the GST pull-down assay. (**K**) GST pull-down assay showing that the K857R mutant of EphA5 SAM gained its binding to SHIP2.

DOI: https://doi.org/10.7554/eLife.35677.005

The following figure supplements are available for figure 2:

**Figure supplement 1.** The structures of the EphA2-SHIP2 and EphA6-Odin SAM–SAM complexes showing the linkers used to connect the SAM domains in both structures are sufficiently long and flexible.

DOI: https://doi.org/10.7554/eLife.35677.006

*Figure 2 continued*

**Figure supplement 2.** The key Arg958[A2]/Phe1226[SHIP2] cation-π interaction was not resolved in the previous NMR study.
DOI: https://doi.org/10.7554/eLife.35677.007
**Figure supplement 3.** Neither EphA2 nor SHIP2 SAM can form homo-oligomers or polymerize into hetero-oligomers.
DOI: https://doi.org/10.7554/eLife.35677.008
**Figure supplement 4.** The critical role of Gly at the beginning of α5 from Eph SAM in SAM–SAM interactions.
DOI: https://doi.org/10.7554/eLife.35677.009
**Figure supplement 5.** Sequence alignments of Eph SAM domains among different family members or across species.
DOI: https://doi.org/10.7554/eLife.35677.010

It is predicted that alteration of any of the interactions in the above-described π-π stacking will perturb the binding of EphA2/6 to SHIP2 or Odin. The most subtle substitution of Arg is probably by the positively charged Lys. Consistent with the binding pattern, the corresponding residue of Arg958 in EphA2 is also found in EphA1 and A6, which could interact with SHIP2 and Odin. Whereas the rest of Eph SAMs contain a Lys in this position except EphA10, which is an Ala (*Figure 2—figure supplement 5*). Based on the structures shown in *Figure 2F*, replacing Arg958[A2] with Lys would eliminate the planar π-system formed by Arg958[A2] and the Ile917[A2]-Lys918[A2] peptide bond and thus seriously weaken the binding, even though the positive charge at the site is retained. Totally consistent with this structural analysis, substitution of Arg958[A2] with Lys completely eliminated the bindings of EphA2 SAM to SHIP2 SAM or Odin SAM (*Figure 2H and I*). Correspondingly, substitution of Phe1226 of SHIP2 SAM with non-aromatic hydrophobic residues (e.g. Leu or Ala) also totally eliminated the binding between SHIP2 SAM and EphA2 SAM (*Figure 2H and J*). Additionally, replacing Asp1222 in SHIP2 SAM with Ala also eliminated the binding between SHIP2 and EphA2, highlighting the importance of the hydrogen bond between Asp1222[SHIP2] and Arg958[A2] in stabilizing the guanidinium group of Arg958 (*Figure 2H*). We mined COSMIC cancer somatic mutation database (*Forbes et al., 2017*; http://cancer.sanger.ac.uk/cosmic) and found that an Arg957 (corresponding to mouse Arg958) to Cys somatic mutation of EphA2 has been detected in ovary carcinoma patients. As expected, substitution of Arg with Cys also completely eliminated EphA2 SAM's binding to the SHIP2 or Odin (*Figure 2H and I*).

The above structural and biochemical analysis highlights the critical role and the exquisite selectivity of Arg958 in EphA2 (or the corresponding Arg in EphA1/EphA6) in terms of bindings to the effectors such as SHIP2 and Odin. We pushed this concept further by testing whether we might be able to convert a non SHIP2/Odin binding Eph SAM domain into a binding one by simply substituting the Lys at the position corresponding Arg958[A2] to Arg (i.e. a single residue 'gain-of-function' mutation). We chose EphA5 SAM domain to test this hypothesis, as it shares ~50% sequence identity to EphA2 SAM but has no or minimal binding to SHIP2 or Odin (*Figure 2H*). Satisfyingly, substituting Lys857 (corresponding Arg958[A2]) with Arg converted EphA5 SAM into a SHIP2 binding SAM domain, though the binding was still relatively weak. The same substitution also enhanced EphA5 SAM's binding to Odin SAM1 by more than 100-fold (*Figure 2H*, *bottom*). Taken together, the above structural and biochemical analysis highlighted that the SAM–SAM domain interactions between Eph receptors and their cytoplasmic effectors are highly specific.

## Specific SAM domain-mediated effector binding is required for ligand-induced cell repulsion function of EphA2

We next tested the cellular function of the specific SAM domain-mediated effector association of the EphA2. DU145 cells, which have a low expression level of endogenous EphA2 receptors, were used to study its ligand-induced inhibition of cell spreading (*Barquilla et al., 2016*; *Lee et al., 2012*; *Miao et al., 2000*; *Shi et al., 2017*). To focus our studies on the forward signal pathway of EphA2, we chose to engage the EphA2 receptor by adding soluble ephrinA1-Fc chimera to the cell culture media. To minimize potential trans signaling from neighboring cells, all experiments were performed in low-density cultures (at about 30% confluency or below). Cells were infected with the lenti-virus containing the 3 × Flag tagged EphA2 FL WT, EphA2-R958K, EphA2 with the SAM domain deleted (EphA2 delSAM), and EphA2 with its SAM substituted by EphA5 SAM (EphA2-SAM$_{A5}$ chimera). Cells infected with the 3 × Flag tagged GFP were the negative controls. As expected and consistent with the previous reports (*Barquilla et al., 2016*; *Shi et al., 2017*), in response to ephrinA1-Fc

stimulation, cells expressing the wild-type full-length EphA2 receptor dramatically retracted and became rounded, whereas the negative control cells did not respond to ephrinA1 (*Figure 3A* and quantified in *Figure 3B*). Using this specific EphA2-mediated cell rounding assay, the repulsion phenotype of the cells expressing EphA2 delSAM, EphA2 R958K or EphA2-SAM$_{A5}$ chimera (each expressed at a similar level with that of EphA2 WT; *Figure 3C*) were evaluated. None of them displayed ephrinA1-induced cell rounding (*Figure 3A and B*), indicating that the highly specific EphA2 SAM domain-mediated effector association is required for the cell spreading function of EphA2.

We next performed cell spreading assay by transfecting SHIP2-SAM to cells infected with lentivirus expressing EphA2 or the GFP control. The data showed that overexpressing SHIP2-SAM effectively reversed the cell collapse phenotype caused by ephrinA1 activation of EphA2, whereas over-

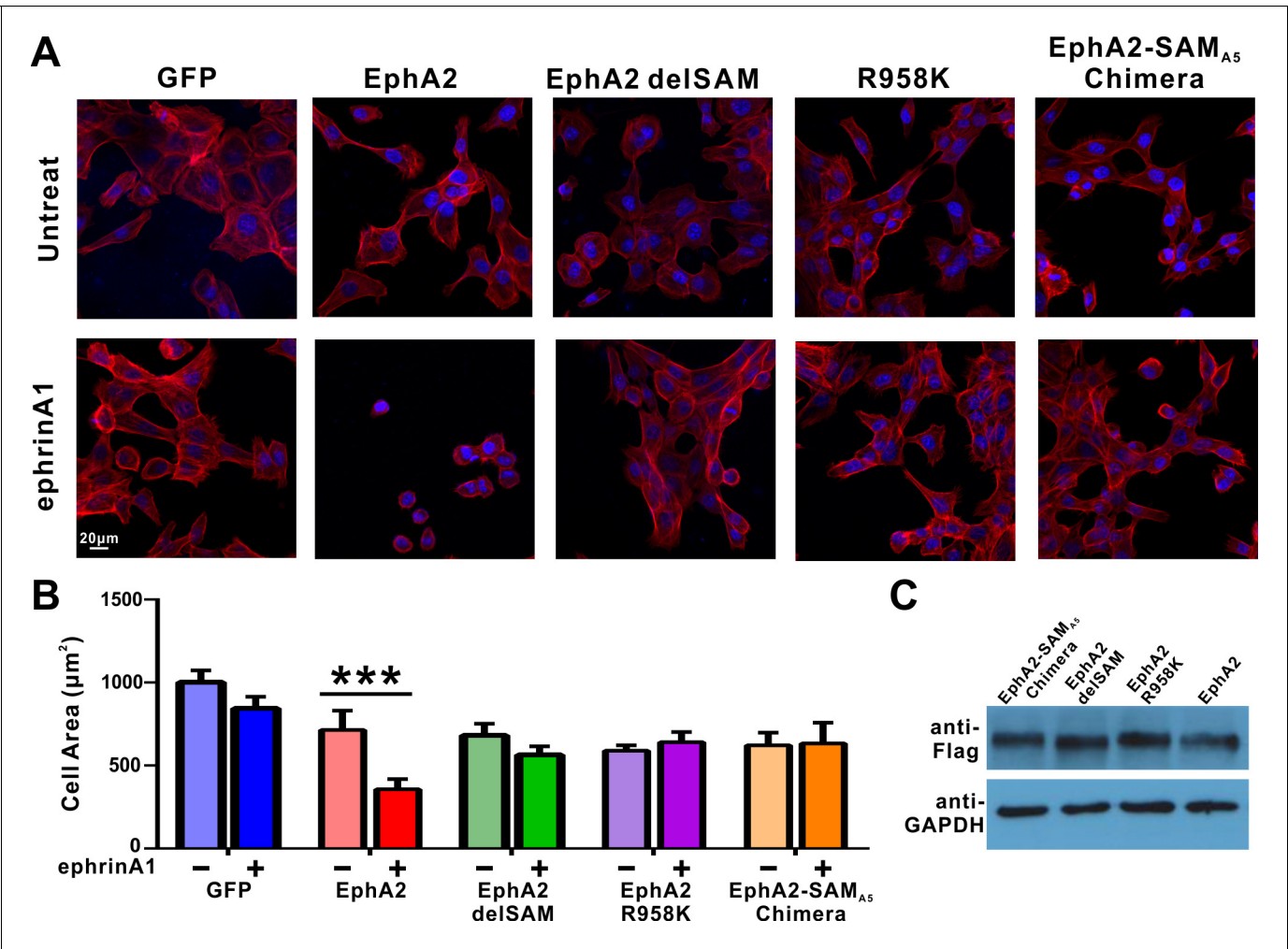

**Figure 3.** Impacts of the mutations in SAM domain of EphA2 on the RTK's function in cell spreading in DU145 cells. (**A**) Selected images showing the morphologies of DU145 cells expressing the wild-type or various mutants of EphA2 before and after ephrin A1-Fc treated. Cells were stained for F-actin with fluorescent phallodin (red), and nuclei were labelled with DAPI (blue). (**B**) Quantification of the cell areas of DU145 cells expressing various constructs. Data represents the mean ±SEM from four independent experiments with each experiment of at least 400 cells (\*\*\*, p<0.001 by Two-way ANOVA with multiple comparisons test). (**C**) Western blot analysis showing the expression levels of different EphA2 (anti-Flag) constructs.
DOI: https://doi.org/10.7554/eLife.35677.011

The following figure supplements are available for figure 3:

**Figure supplement 1.** SHIP2-SAM effectively blocks the ephrinA1/EphA2-induced cell retraction.
DOI: https://doi.org/10.7554/eLife.35677.012

**Figure supplement 2.** Impacts of the mutations in SAM domain of EphA2 on the RTK's function in cell spreading in HEK293T cells.
DOI: https://doi.org/10.7554/eLife.35677.013

expressing SHIP2-SAM did not lead to any observable changes in the GFP control (*Figure 3—figure supplement 1*), indicating that this blocking effect by SHIP2-SAM is EphA2-dependent.

Arg957 (corresponding to Arg958 of mouse EphA2 SAM), which is found to be mutated to Cys in patients with ovary carcinoma, is located in the SAM-SAM binding interface. We also tested the impact of this mutation as well as the R958K mutation using HEK293T cells, another cell line widely used for cell spreading assay (*Lawrenson et al., 2002*; *Yamazaki et al., 2009*). Similar to the R958K mutation in the DU145 cells assay, we also observed that HEK293T cells expressing either of the EphA2 R958C or R958K mutants lost the capacity to undergo ephrinA1-induced retraction, whereas cells expressing EphA2 WT showed effective ephrinA1-induced retractions (*Figure 3—figure supplement 2*).

## SAMD5 as a new effector of eph SAM

The above crystal structures of EphA2-SHIP2 and EphA6-Odin revealed the importance of the negatively charged residues within the α2-α3 loop and at the beginning of the α4 helix of SAM domain in binding to Eph SAM (*Figure 2* and *Figure 4A*). We tried to look for new binding partners for Eph SAMs. By inspecting each of the 163 SAM domains in 127 mouse SAM domain containing proteins (SMART's nrdb database: http://smart.embl-heidelberg.de/), we found several potential candidates that may interact with Eph SAM. AIDA1, also named as Ankyrin repeat and sterile alpha motif domain-containing protein 1B (Anks1b), is a paralog of Odin (Anks1a) and thus its SAM1 domain is expected to bind to EphA2/A6 SAM with the same mode and specificity as Odin SAM1 does. Another promising candidate is SAMD5, whose SAM domain is similar to SHIP2 SAM (*Figure 4A*) and highly expressed in breast cancer cells and mainly cytoplasmic (*Lo et al., 2015*).

We purified SAMD5 SAM and measured its binding to the SAM domain of every Eph receptor. Indeed, SAMD5 SAM was found to be a strong binder of Eph SAM domains such as EphA5 and

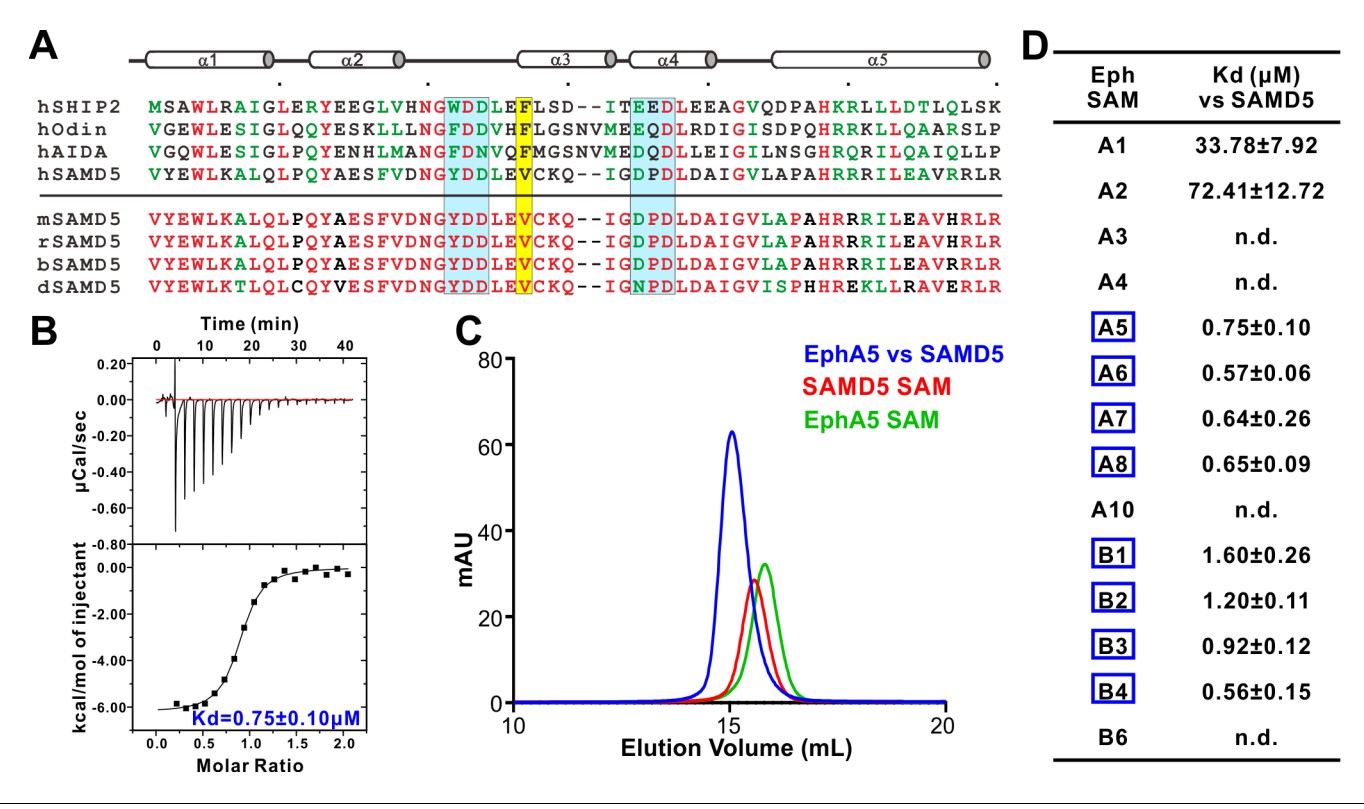

**Figure 4.** Characterization of SAMD5 as a new effector of Eph SAM. (**A**) Sequence alignment of the SAM domains of SHIP2, Odin and two potential interaction partners (AIDA1 and SAMD5). (**B** and **C**) ITC (**B**) and analytical gel-filtration chromatography (**C**) showing that SAMD5 binds to EphA5 SAM. (**D**) ITC-derived dissociation constants showing that SAM domains from EphA5-8 and EphB1-4 bind to SAMD5 SAM.

DOI: https://doi.org/10.7554/eLife.35677.014

others (*Figure 4B–D*). Surprisingly, although the amino acid sequence of SAMD5 SAM can be aligned well (the key binding residues in particular) with the SAM domain sequences of SHIP2 and Odin (*Figure 4A*, *top*), SAMD5 bound to EphA5-8 and EphB1-4 SAM domains with strong affinities but only weakly to EphA1/2 SAM domains (*Figure 4D*). This biochemical data further substantiated that the SAM domains of Eph receptors can provide diverse binding specificities to their effectors.

## EphA5/SAMD5 SAM-SAM complex structure reveals the binding specificity

To delineate the mechanism governing the Eph/SAMD5 interaction, we solved the EphA5/SAMD5 SAM–SAM complex crystal structure at the resolution of 1.9 Å (*Table 1*). Similar to that of EphA2/SHIP2 or EphA6/Odin, EphA5 SAM also uses its End-Helix to interact with the Mid-Loop SAMD5 SAM (*Figure 5A and B*). However, their detailed binding mechanisms are quite different. The most prominent difference is that the EphA5/SAMD5 SAM-SAM does not contain the π-π stacking observed in the EphA2/SHIP2 and EphA6/Odin complexes (*Figure 5C*). Instead, the EphA5/SAMD5 SAM-SAM is essentially completely mediated by charge–charge and hydrogen bonding interactions (*Figure 5C*). Perturbation of these interactions invariably weakened the interaction (*Figure 5E*). Similar to what we observed in the EphA2/SHIP2 and EphA6/Odin structures, Gly853[A5] at the beginning of α5 in EphA5 SAM is also critical for its binding to SAMD5 SAM. The backbone amine of Gly853[A5] forms a strong hydrogen bond with Tyr27[SAMD5] sidechain hydroxyl group (*Figure 5C*). This hydrogen bond brings the α2-α3 loop of SAMD5 very close to α5 of EphA5 (*Figure 2—figure supplement 4C*), and the site has no room to accommodate any residues with side chains, further highlighting the importance of a Gly at this position in the Eph SAM domains (*Figure 2—figure supplement 5A*). To our surprise, substitution of Tyr27[SAMD5] with Phe completely eliminated SAMD5's binding to EphA5 (*Figure 5E*), indicating the critical role of the Gly853[A5]-Tyr27[SAMD5] hydrogen binding in addition to their steric role.

Our crystal structure of the EphA5/SAMD5 complex can explain why SAMD5 does not bind to EphA1-4, EphA10, and EphB6. For the SAM domains from EphA3/EphA4/EphA10/EphB6, each has one or more of the key residues missing (e.g. EphA10 and EphB6 are missing Gly at the beginning of α5, and EphA3/4 SAMs are missing positively charged Lys at the end of α2 or in the middle of α5; *Figure 2—figure supplement 5*). For the EphA1/EphA2 SAM domains, although all the key residues are present, it is noticed that a Pro residue is at the position corresponding to Val852[A5] (*Figure 2—figure supplement 5*). The sidechain pyrrolidine ring of Pro would introduce steric hindrance with Asp40, and thus may perturb SAMD5 from binding to EphA1/EphA2 SAM domains (*Figure 5D*), resulting in low affinities (*Figure 4D*). Supporting this analysis, replacing Val825 of EphA5 SAM with a Pro led to about 10-fold affinity decrease in its binding to SAMD5. Additionally, substitution Pro953 in EphA2 with a smaller residue Ala increased its SAMDs SAM binding by more than 10 fold (*Figure 5E*). Finally, EphA6 SAM could bind to both SAMD5 and SHIP2/Odin SAM domains (*Figure 1D and 4D*), since it satisfies the binding criteria of both types.

## Disease mutations of Eph SAM domains

Eph receptors mediate cell-cell contact signaling and are vital for cell adhesion and migration (*Kania and Klein, 2016*). It is not surprising that mutations of Eph receptors can cause various human diseases including cancers (*Barquilla and Pasquale, 2015*; *Boyd et al., 2014*; *Gaitanos et al., 2015*; *Genander and Frisén, 2010*; *Kania and Klein, 2016*; *Pasquale, 2005*; *2008*; *2010*). We surveyed the COSMIC cancer somatic mutation database (*Forbes et al., 2017*; http://cancer.sanger.ac.uk/cosmic; a collection of somatic mutations found in cancer patients mainly from large-scale genome sequencing studies) and found that numerous mutations occur in almost every subtype of Eph genes except for EphB6. A significant number of these mutations fell into the SAM domain regions (*Figure 6A*). The biochemical and structural information provided in this study allowed us to test the impact of these mutations found in cancer patients in terms of binding to the cytoplasmic effectors including SHIP2 and SAMD5. For the practical workload reason, we chose to investigate the mutation sites that are at the positions critical for the SAM/SAM interactions analyzed in our structural studies shown in *Figure 2 and 5*, and are shaded in orange in *Figure 6A*. We purified each of these mutant SAM domains of Eph proteins, and measured their individual bindings to SHIP2 SAM or SAMD5 SAM by ITC-based assays. The results are summarized in *Figure 6B and C*.

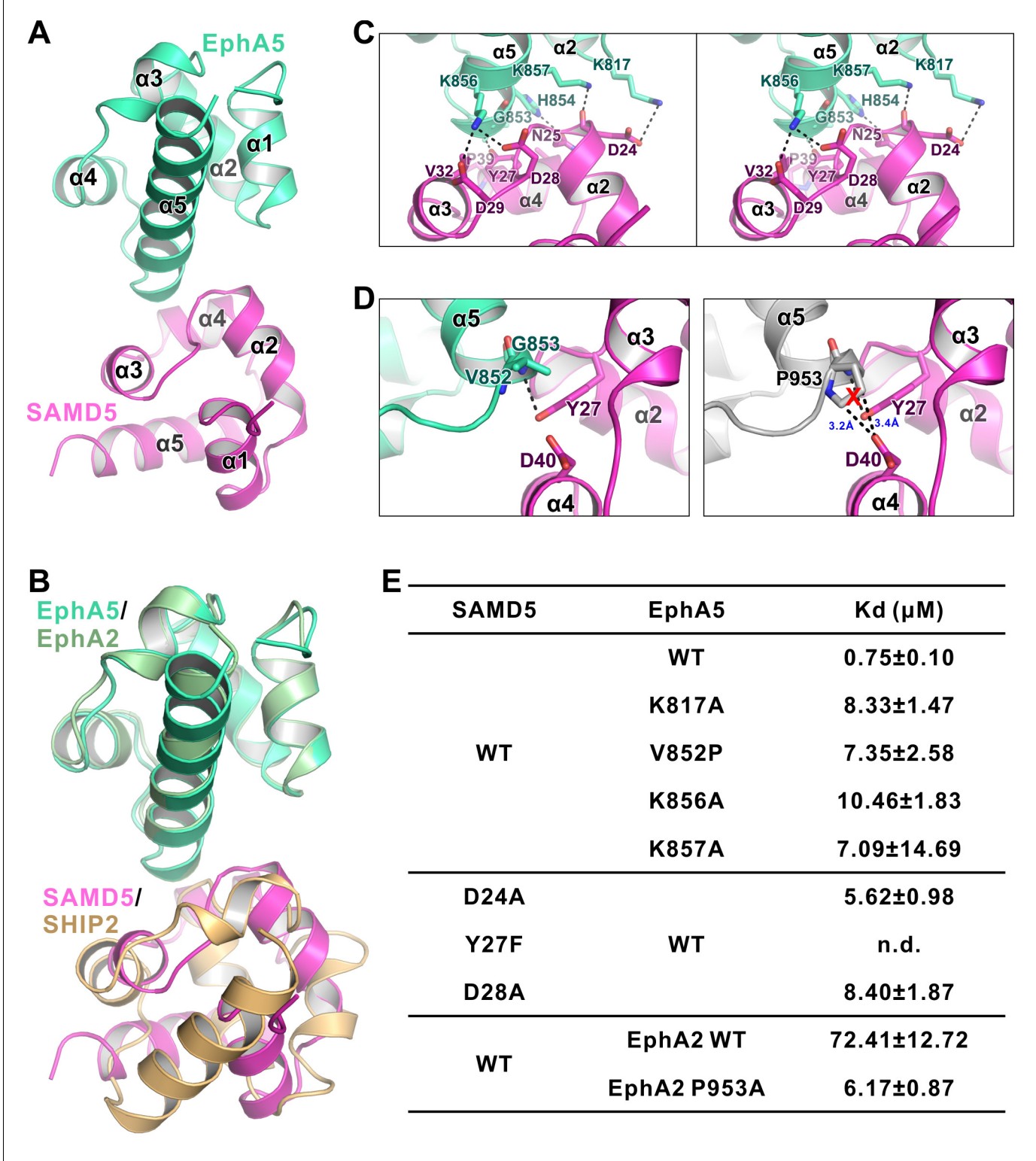

**Figure 5.** The detailed interaction between EphA5 SAM and SAMD5 SAM. (**A**) Ribbon diagram showing the overall structure of the EphA5/SAMD5 complex. (**B**) Superposition of the EphA5/SAMD5 and EphA2/SHIP2 complex structures by aligning EphA5 and EphA2 together. SAMD5 and SHIP2 use different surfaces to interact with the same site on the SAM domains from EphA5 and EphA2. (**C**) Stereo view of the detailed interaction of the EphA5/SAMD5 complex. Key residues are shown with the stick model. Salt bridges and hydrogen bonds are indicated with dashed lines. (**D**) Structure model showing that V852P mutation of EphA5 will introduce steric hindrance with Tyr27 and Asp40 from SAMD5 and thus impair the binding. (**E**) ITC-derived

*Figure 5 continued on next page*

*Figure 5 continued*
dissociation constants showing mutations of critical residues weaken or abolish the binding between SAMD5 and EphA5, and a gain-of-function mutation of EphA2 SAM (P953A) with enhanced binding to SAMD5.
DOI: https://doi.org/10.7554/eLife.35677.015

As expected, EphA1 R966C, EphA2 R957C, and EphA6 R1014Q mutations (the cation-π forming Arg at α5 helix) totally abolished their binding to SHIP2 (*Figure 6B*). Similar deleterious effect can also be seen in EphA1 α1-α2 loop mutation R926G (*Figure 6B*). EphA5 G1014S, EphA7 G972V, and EphB3 G974D mutations disrupted their binding to SAMD5 (*Figure 6C*), further supporting our structural finding that the sidechain-less Gly at the beginning of α5 is critical (*Figure 2—figure*

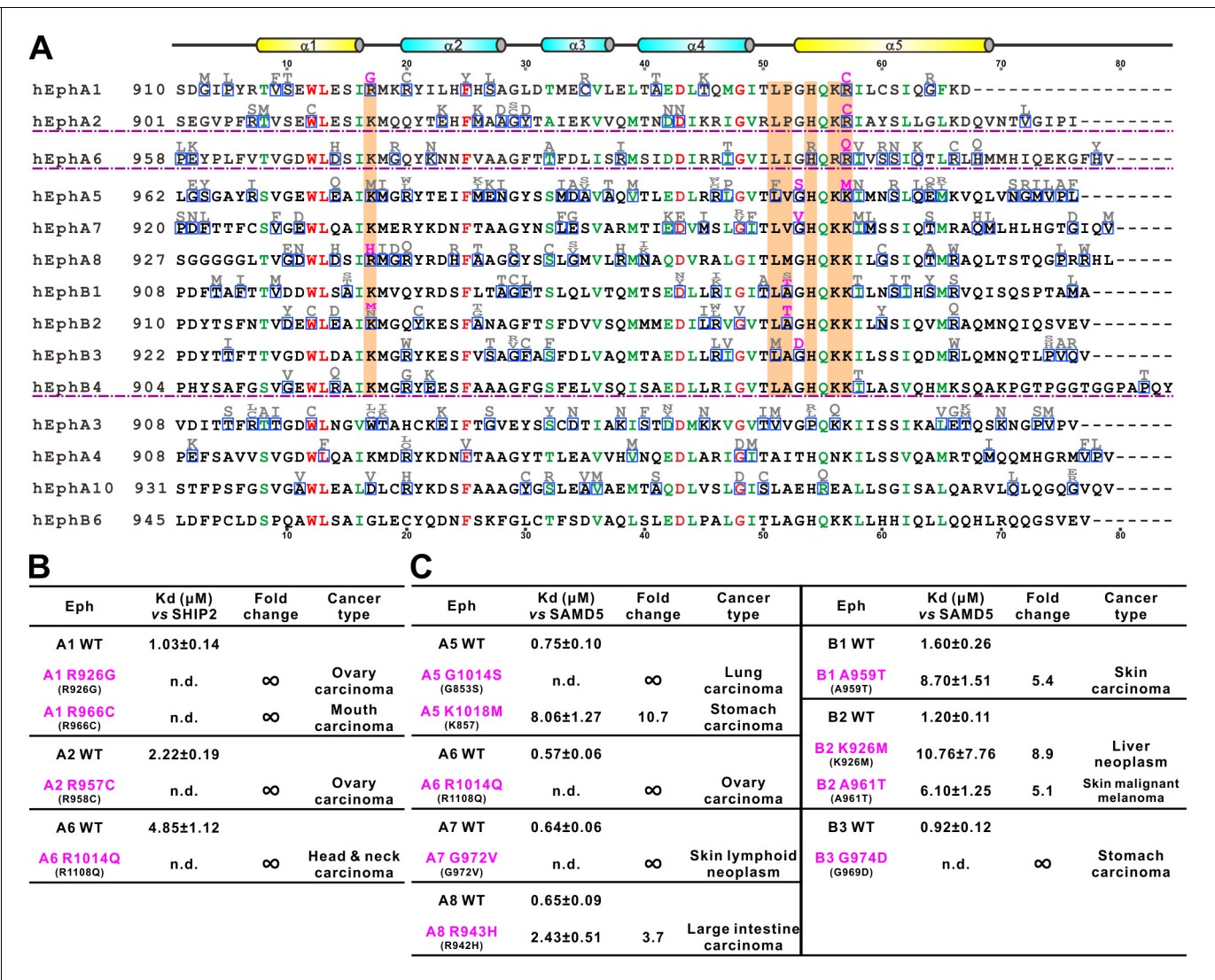

**Figure 6.** Disease mutations of Eph SAM domains. (**A**) Sequence alignments of Eph SAMs. The secondary structure elements are labeled according to the EphA2 SAM structure. Residues that are identical and highly similar are shown in red and green, respectively. The residues critical in the SAM–SAM interactions are shaded in orange. Mutations within the SAM domain of all Eph receptors found in cancer patients are indicated in grey or magenta. The residues indicated in magenta were experimentally tested and shown in B and C. (**B**) Summary of the impacts of selected mutations of EphA1, A2, A6 identified in cancer patients (see panel A) on the SHIP2 binding. (**C**) Summary the impacts of selected mutations of EphA5, A6, A7, A8 and EphB1, B2, B3 identified in cancer patients (see panel A) on the SAMD5 binding.
DOI: https://doi.org/10.7554/eLife.35677.016

*supplement 4*). Mutations on other critical sites such as EphA5 K1018M, EphA6 R1014Q, EphA8 R943H, and EphB1 K926M, A961T all weakened or even disrupted their binding to SAMD5 (*Figure 6C*). Taken together, the above analysis of the impact of the Eph SAM mutations found in cancer patients illustrated the tremendous values of the structural and biochemical studies of Eph SAM-mediated target interactions presented in the current study.

## Discussion

Since the discovery of the Eph receptors three decades ago, numerous efforts have been invested in elucidating ligand-induced signaling mechanisms of this classical family of RTKs in many biological processes. However, in contrast to the explosive amount of information on the functions uncovered, our understandings of the forward signaling mechanisms of the receptors are relatively poor. It is particularly unclear how Eph receptors may engage their specific cytoplasmic effectors to transmit biological signals using their highly similar cytoplasmic domains. At present, very little knowledge is available on what downstream targets of Eph receptors are and whether each Eph receptor engages different cytoplasmic effectors upon ligand binding, and this has seriously hampered our understanding on the action mechanisms of this family of RTKs in both physiological and patho-physiological conditions.

In this study, we performed systematic biochemical and structural studies of the bindings of three SAM domains, two from previously identified Eph-binding proteins (SHIP2 and Odin) and one from a new Eph-binding protein discovered in this study (SAMD5), to the SAM domains from every Eph receptor. Our study reveals that the Eph SAM domains have exquisitely specific target SAM domain-binding properties, although their overall SAM-SAM heterodimer formation modes are very similar. The binding affinities for all the interactions reported here are in the submicromolar to micromolar range. In living cells, their binding avidities are likely to be further enhanced due to the following two reasons. First, Eph receptors are known to multimerize upon ligand activation, mainly via their extracellular domains (*Janes et al., 2012*; *Seiradake et al., 2013*). The multimerization of the extracellular domains can bring a number of Eph cytoplasmic tails in close proximity. Second, several studies have revealed that effectors like SHIP2 can be recruited to the cell membrane via SH2 domain mediated interactions to other cell surface receptors (*Pesesse et al., 2001*; *Wang et al., 2004*). These two mechanisms can increase the local concentrations of Ephs and their downstream targets, and thus enhance their binding avidities. This result provides a partial answer in rationalizing diverse functions of Eph receptors that can be induced by the same or similar ephrin ligand(s). It is anticipated that there exist additional SAM domain-containing proteins capable of binding to Eph receptors. Since there are 110 SAM domain-containing proteins in addition to the 14 Eph receptors and three Eph-binding proteins studied here, one approach to identify new Eph SAM-binding proteins is to survey bindings of all 14 Eph SAM domains to the rest of the SAM domains in the human proteome using purified recombinant proteins by protein array-based methods.

We demonstrated that, unlike many other previously characterized polymer forming SAM domains, every Eph receptor SAM domain adopts stable monomers in solution (*Harada et al., 2008*; *Knight et al., 2011*; *Stapleton et al., 1999*; *Thanos et al., 1999*; *Wang et al., 2016*). Additionally, no SAM–SAM hetero-complexes between Eph SAM domains could form (data not shown). The structures solved in this study revealed that the SAM domains from EphA2, EphA5, and EphA6 all bind to its effector SAM domain using their respective End-Helix. It is noticed that every Eph SAM domain also contains an extremely conserved Mid-Loop (see EphA2/EphA5/EphA6 SAM sequence alignments in *Figure 2—figure supplement 5B–D* for an example). It is compelling to speculate that at least some Eph SAM domains may also use their Mid-Loops to bind to the End-Helix of their effector SAM domains. This hypothesis is further supported by the observation that many residues in the Mid-Loop regions of Eph SAM domains are found to be mutated in cancer patients (*Figure 6A*).

The structures of Eph SAM domains in complex with the SAM domains of several different effectors presented in this study are very useful in interpreting numerous mutations/variants found in the Eph SAM domains in patients with different diseases (e.g. cancers analyzed in *Figure 6*). It is expected that not all mutations occur in the SAM domains will alter functions of Eph receptors. The structures of the SAM–SAM complexes presented in this work, coupled with amino acid sequence-based analysis, readily predict the following three categories of mutations that will likely alter

functions of Eph receptors via changing their SAM domain structures and effector binding. First, mutations of residues in the folding core or residues playing other critical structural roles (e.g. residues highlighted in red in *Figure 6A*) may impair the overall structure and thus effector binding of Eph SAM domains. We tested a few of such mutations (e.g. EphA2 W913C and D944N), and found that these SAM mutants invariably expressed as inclusion bodies (data not shown). Second, mutations occur in the End-Helix region of Eph SAM domains. This category of Eph SAM domain mutants often has defects or even total impairments in binding to their effectors (*Figure 6B and C*), and thus are expected to have impaired downstream signaling. Third, mutations occur in the Mid-Loop region of the Eph SAM domains may also impair their binding to effector SAM domain as we mentioned above, although this prediction will need to be verified. We propose that the Eph SAM mutations found in patients that fall into the above three categories will have higher chance to be disease-relevant mutations and thus are given higher priority to be investigated. Certainly, it cannot be ruled out that mutations of the Eph SAM domains that are outside the three categories may also impair functions of Eph receptors, perhaps via still unknown SAM domain-mediated target bindings.

# Materials and methods

## Key resources table

| Reagent type (species) or resource | Designation | Source or reference | Identifiers |
|---|---|---|---|
| Strain, strain background (E. coli) | BL21(DE3) | Invitrogen | Cat #C600003 |
| Cell line (Human) | DU145 | ATCC | Cat #HTB-81; RRID:CVCL_0105 |
| Cell line (Human) | HEK293T | ATCC | Cat #CRL-3216; RRID:CVCL_0063 |
| Transfected construct (Lenti-virus) | pLVX-CMV-EGFP-3 ×Flag-CoPuroR | Shanghai Taitool Bioscience Co., Ltd | NA |
| Transfected construct (Lenti-virus) | pLVX-CMV-EphA2 −3 × Flag-CoPuroR | Shanghai Taitool Bioscience Co., Ltd | NA |
| Transfected construct (Lenti-virus) | pLVX-CMV-EphA2 -R958K-3×Flag-CoPuroR | Shanghai Taitool Bioscience Co., Ltd | NA |
| Transfected construct (Lenti-virus) | pLVX-CMV-EphA2 delSAM-3 × Flag-CoPuroR | Shanghai Taitool Bioscience Co., Ltd | NA |
| Transfected construct (Lenti-virus) | pLVX-CMV-EphA2-SAM$_{A5}$−3 × Flag-CoPuroR | Shanghai Taitool Bioscience Co., Ltd | NA |
| Transfected construct (plasmid) | GFP-EphA2 | This paper | NA |
| Transfected construct (plasmid) | GFP-EphA2-R958K | This paper | NA |
| Transfected construct (plasmid) | GFP-EphA2-R958C | This paper | NA |
| Transfected construct (plasmid) | Myc-SHIP2-SAM | This paper | NA |
| Antibody | monoclonal Anti-Flag M2, clone M2 | Sigma | Cat #F3165; RRID:AB_259529 |
| Antibody | Anti-Myc (9B11) mouse mAb | Cell Signaling Technology | Cat #2276; RRID: AB_331783 |
| Antibody | Anti-EphA2 (D4A2) XP Rabbit mAb | Cell Signaling Technology | Cat #6997; RRID:AB_10827743 |
| Antibody | Anti-GAPDH (D16H11) XP Rabbit mAb | Cell Signaling Technology | Cat #5174; RRID:AB_10622025 |
| Peptide, recombinant protein | Recombinant Mouse Ephrin-A1-Fc Chimera | R and D Systems | Cat #602-A1 |

*Continued on next page*

*Continued*

| Reagent type (species) or resource | Designation | Source or reference | Identifiers |
|---|---|---|---|
| Commercial assay or kit | Clone Express II, One-Step Cloning Kit | Vazyme Biotech Co., Ltd | Cat #C112 |
| Commercial assay or kit | ViaFect transfection reagent | Promega Corporation | Cat #E4981 |
| Chemical compound, drug | Rhodamine Phalloidin | Cytoskeleton Inc. | Cat #PHDR1 |
| Software, algorithm | Origin7.0 | OriginLab | http://www.originlab.com/; RRID: SCR_002815 |
| Software, algorithm | GraphPad Prism | GraphPad Software Inc. | http://www.graphpad.com/scientific-software/prism/; RRID: SCR_002798 |
| Software, algorithm | HKL2000 | HKL Research Inc. | http://www.hkl-xray.com/ |
| Software, algorithm | CCP4 | PMID: 21460441 | http://www.ccp4.ac.uk/; RRID: SCR_007255 |
| Software, algorithm | PHENIX | PMID: 20124702 | http://www.phenix-online.org/; RRID: SCR_014224 |
| Software, algorithm | PyMOL | DeLano Scientific LLC | http://www.pymol.org/; RRID: SCR_000305 |
| Software, algorithm | ASTRA6.1 | Wyatt Technology Corporation | http://www.wyatt.com/products/software/astra.html |
| Software, algorithm | ImageJ | NIH | https://imagej.nih.gov/ij/; RRID: SCR_003070 |
| Other | DAPI | Sigma | Cat #D9542 |

## Protein expression and purification

DNA encoding EphA2 SAM (NP_034269.2, residues 901–977), SHIP2 SAM (NP_034697.2, residues 1201–1257), Odin SAM1 (NP_852078.1, residues 712–776) and SAMD5 SAM (NP_796245.2, residues 1–66) were amplified from *Mus musculus* cDNA libraries as the template and individually cloned into a modified pET vector (*Liu et al., 2011*). All mutants were created using the standard two-step PCR methods. The fusion constructs of SHIP2 SAM/EphA2 SAM and Odin SAM1/EphA6 SAM each contained a TEV protease recognition site 'ENLYFQ' flanked by several Gly-Ser repeats as the flexible linkers.

Recombinant proteins with N-terminal $His_6$-tag were expressed *E.coli* BL21 (DE3) strain. Expressed proteins were purified by the $Ni^{2+}$-NTA Sepharose 6 Fast Flow beads (GE Healthcare, China) affinity chromatography followed by a Superdex-200 prep grade size-exclusion chromatography (GE Healthcare) in a buffer containing 50 mM Tris, pH 7.5, 100 mM NaCl, 1 mM DTT, and 1 mM EDTA. The N-terminal His-tag of each protein was cleaved by 3C protease and removed by another step of size-exclusion chromatography.

## Analytical gel-filtration chromatography

Analytical gel-filtration chromatography was performed on an AKTA system (GE Healthcare) using the Superose 12 10/300 GL column. Protein samples (each with 100 μL at 50 μM) were injected into the column pre-equilibrated with a buffer containing 50 mM Tris, pH 7.5, 100 mM NaCl, 1 mM DTT, and 1 mM EDTA.

## Isothermal titration calorimetry assay

Isothermal titration calorimetry experiments were carried out on the MicroCal ITC200 calorimeter (Malvern, UK) at 25°C. The concentration of the injected samples in the syringe was 500 μM, and the concentration of the samples in the cell was fixed at 50 μM. The sample in the syringe was sequentially injected into the sample cell with a time interval of 150 s (0.5 μL for the first injection and 2 μL each for the following 19 injections). The titration data were analyzed by the Origin 7.0 software and fitted with the one-site binding model.

## Protein crystallography

The SHIP2 SAM-EphA2 SAM fusion protein (with a 14-residue linker 'SSGENLYFQSGSSG'), the Odin SAM-EphA6 SAM fusion protein (with a 17-residue linker 'PSGSSGENLYFQSGSSG'), and a 1:1 mixture of SAMD5 SAM and EphA5 SAM were concentrated to ~10–20 mg/mL for crystallization. Crystals were obtained at 16°C by the sitting-drop vapor diffusion against 80 μL well solution using 48-well format crystallization plates. SHIP2 SAM-EphA2 SAM complex crystals were grown in a buffer containing 0.2 M Succinic acid pH 7.0, 20% w/v Polyethylene glycol 3350. Odin SAM1-EphA6 complex crystals were grown in a buffer containing 0.1 M HEPES sodium pH 7.5, 1.5 M lithium sulfate monohydrate. SAMD5 SAM-EphA5 SAM complex crystals were grown in a buffer containing 0.2 M ammonium acetate, 0.1 M BIS-TRIS pH 5.5, 25% w/v Polyethylene glycol 3350. Crystals were soaked in the crystallization solution containing additional 5% (for SHIP2/EphA2 and SAMD5/EphA5) or 20% (for Odin/EphA6) v/v glycerol for cryo-protection. Diffraction data were collected at the Shanghai Synchrotron Radiation Facility BL17U1 at 100 K. Data were processed and scaled using HKL2000 (*Otwinowski and Minor, 1997*).

Structures were all solved by molecular replacement with the EphA2 SAM domain (PDB: 3KKA) and the SHIP2 SAM domain (PDB: 2K4P) or the Odin first SAM domain (PDB: 2LMR) structures as the searching models using PHASER (*McCoy et al., 2007*). Further manual model buildings and refinements were completed iteratively using Coot (*Emsley et al., 2010*) and PHENIX (*Adams et al., 2010*) or Refmac5 (*Murshudov et al., 2011*). The final models were validated by MolProbity (*Chen et al., 2010*). The final refinement statistics are summarized in *Table 1*. The structure figures were prepared by PyMOL (http://www.pymol.org). The structure factors and the coordinates of the structures reported in this work have been deposited to PDB under the accession codes of 5ZRX, 5ZRY and 5ZRZ for the EphA2/SHIP2, EphA6/Odin and EphA5/SAMD5 complex structures, respectively.

## GST pulldown assays

GST-tagged EphA2, EphA5, SHIP2, SAMD5 SAM domains and their mutant proteins or GST alone were incubated with HEK293T cell lysates expressing the GFP-tagged target proteins for 1 hr at 4°C. The mixture was then loaded onto 20 μL Glutathione Sepharose 4B beads (GE Healthcare) in PBS buffer for 0.5 hr at 4°C. After washing twice, the proteins captured by the beads were eluted by boiling with SDS-PAGE loading buffer, resolved by SDS-PAGE and detected by an anti-GFP antibody using western blotting.

## Cell spreading assay

Both DU145 and HEK293T cells were cultured in Dulbecco's Modified Eagle Medium (DMEM) supplemented with 10% fetal bovine serum (FBS), and 1% of penicillin-streptomycin at 37°C with 5% $CO_2$. HEK293T cells were transfected with GFP-tagged full-length wild type EphA2, EphA2-R958K, EphA2-R958C and GFP alone using ViaFect Transfection Reagent (Promega, Madison, WI) as per manufacturer's protocol. The full-length wild type EphA2, EphA2-R958K, EphA2 delSAM, EphA2-SAM$_{A5}$ chimera and GFP control, each with a C-terminal 3 × Flag tag, were individually cloned into the pLVX plasmid for commercial viral packaging (Shanghai Taitool Bioscience Co. Ltd, China). All infectious lenti-viruses were produced in HEK293T cells by co-transfected with the VSV-G glycoprotein expressing plasmid pMD2.G and pCMVΔR8.91 packaging construct. The lenti-viruses were used to infect DU145 cells and selected with 1 μg/mL puromycin for ten days. For experiments involving SHIP2-SAM, DU145 cells infected with lenti-virus expressing GFP control or full-length wild type EphA2 were further transfected with Myc-tagged SHIP2-SAM using ViaFect Transfection Reagent (Promega). The expression levels of proteins were verified by immunoblotting. These cells were not individually authenticated and not found to be on the list of commonly misidentified cell lines (International Cell Line Authentication Committee). Cells were tested negative for mycoplasma contamination by cytoplasmic DAPI staining.

Cells were preserved to passage into a chamber (ibid μ-Slide VI$^{0.4}$, Germany), starved for 4 hr in serum-free medium, and then stimulated with ephrinA1-Fc (R and D Systems, 1 μg/mL in PBS) for 30 min at 37°C. The cells were then fixed for 15 min with 4% paraformaldehyde in PBS and stained with rhodamine-conjugated phalloidin for 30 min. Cell morphologies were analyzed with a Zeiss Confocal microscope (LSM710) or an Olympus fluorescence microscope (BX61) equipped with a digital

camera with a 20 × objective lens. The cell areas were measured using the ImageJ software (https://imagej.nih.gov/ij/). Two-way ANOVA with multiple comparisons test was used to compare cell areas among different experimental groups, and presented with the GraphPad Prism software.

## Acknowledgements

We thank the Shanghai Synchrotron Radiation Facility (SSRF) BL17U1 and BL19U1 for X-ray beam time. This work was supported by grants from the Minister of Science and Technology of China (2014CB910204), National Key R and D Program of China (2016YFA0501903), Natural Science Foundation of Guangdong Province (2016A030312016), Shenzhen Basic Research Grant (JCYJ20160229153100269) and Asia Fund for Cancer Research to MZ, and grants from the National Natural Science Foundation of China (No. 31670765) and Shenzhen Basic Research Grants (JCYJ20160427185712266 and JCYJ20170411090807530) to WL MZ is a Kerry Holdings Professor in Science and a Senior Fellow of IAS at HKUST.

## Additional information

### Competing interests

Mingjie Zhang: Reviewing editor, *eLife*. The other authors declare that no competing interests exist.

### Funding

| Funder | Grant reference number | Author |
| --- | --- | --- |
| Ministry of Science and Technology | 2014CB910204 | Mingjie Zhang |
| Natural Science Foundation of Guangdong Province | 2016A030312016 | Mingjie Zhang |
| Shenzhen Basic Research Grant, Shenzhen, China | JCYJ20160229153100269 | Wei Liu |
| National Natural Science Foundation of China | 31670765 | Wei Liu |
| Asia Fund for Cancer Research | AFCR17SC01 | Mingjie Zhang |
| Ministry of Science and Technology | 2016YFA0501903 | Mingjie Zhang |
| Shenzhen Basic Research Grant, Shenzhen, China | JCYJ20160427185712266 | Wei Liu |
| Shenzhen Basic Research Grant, Shenzhen, China | JCYJ20170411090807530 | Wei Liu |

The funders had no role in study design, data collection and interpretation, or the decision to submit the work for publication.

### Author contributions

Yue Wang, Validation, Investigation, Methodology, Writing—original draft; Yuan Shang, Investigation, Methodology, Writing—original draft; Jianchao Li, Investigation, Methodology, Writing—original draft, Writing—review and editing; Weidi Chen, Investigation; Gang Li, Jun Wan, Writing—original draft, Project administration; Wei Liu, Formal analysis, Supervision, Writing—original draft, Project administration; Mingjie Zhang, Conceptualization, Supervision, Funding acquisition, Writing—original draft, Project administration, Writing—review and editing

### Author ORCIDs

Jianchao Li http://orcid.org/0000-0002-8921-1626
Wei Liu https://orcid.org/0000-0001-8250-2562
Mingjie Zhang http://orcid.org/0000-0001-9404-0190

Decision letter and Author response
Decision letter https://doi.org/10.7554/eLife.35677.029
Author response https://doi.org/10.7554/eLife.35677.030

## Additional files

### Supplementary files

• Transparent reporting form
DOI: https://doi.org/10.7554/eLife.35677.017

### Data availability

The structure factors and the coordinates of the structures reported in this work have been deposited to PDB under the accession codes of 5ZRX, 5ZRY and 5ZRZ for the EphA2/SHIP2, EphA6/Odin and EphA5/SAMD5 complex structures, respectively.

The following datasets were generated:

| Author(s) | Year | Dataset title | Dataset URL | Database, license, and accessibility information |
|---|---|---|---|---|
| Zhang M | 2018 | Specific Eph receptor-cytoplasmic effector signaling mediated by SAM-SAM domain interactions | https://www.rcsb.org/structure/5ZRX | Publicly available at the RCSB Protein Data Bank (accession no: 5ZRX) |
| Zhang M | 2018 | Data from: Specific Eph receptor-cytoplasmic effector signaling mediated by SAM-SAM domain interactions | https://www.rcsb.org/structure/5ZRY | Publicly available at the RCSB Protein Data Bank (accession no: 5ZRY) |
| Zhang M | 2018 | Data from: Specific Eph receptor-cytoplasmic effector signaling mediated by SAM-SAM domain interactions | https://www.rcsb.org/structure/5ZRZ | Publicly available at the RCSB Protein Data Bank (accession no: 5ZRZ) |

The following previously published dataset was used:

| Author(s) | Year | Dataset title | Dataset URL | Database, license, and accessibility information |
|---|---|---|---|---|
| Forbes SA | 2017 | COSMIC | http://cancer.sanger.ac.uk/cosmic | Publicly available at the Cosmic website (the Catalogue Of Somatic Mutations In Cancer) |

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
