## [Decision Letter]

Thank you for submitting your article "Specific Eph Receptor-Cytoplasmic Effector Signaling Mediated by SAM-SAM Domain Interactions" for consideration by *eLife*. Your article has been favorably evaluated by John Kuriyan (Senior Editor) and three reviewers, one of whom, Volker Dötsch (Reviewer #1), is a member of our Board of Reviewing Editors.

The reviewers have discussed the reviews with one another and the Reviewing Editor has drafted this decision to help you prepare a revised submission.

Summary:

The manuscript gives in great detail a systematic analysis of the interactions of the SAM domains of EphA (and EphB) receptors with other known SAM-domain containing proteins (ODIN and SHIP2) and identify a new SAM-domain containing protein interacting with some of the EphA SAM domains.

The paper is interesting since it further describes in very structural detail the specificity in the interaction between these SAM proteins and various EphA and EphB proteins, which will be helpful in the future to further decipher/understand the differential activities of individual EphAs after activation by ephrinAs, which is at present difficult to explain given the highly similar intracellular domains including, for example, the promiscuity of the COOH terminal PDZ motifs.

Essential revisions:

1) PDB submission codes should be provided.

2) A binding constant of 2.2 μM is still rather weak considering cellular concentrations. In principle such weak binding can be functional through multimerization effects (although no multimerization seems to occur with the investigated SAM domains) or higher local concentrations in specific compartments (such as near the membrane). This should be discussed.

3) Related to this, is there any evidence that SAM domains of different Eph receptors can mediate receptor clustering / heterocomplex formation?

4) It would be interesting to study functionally the role of the SAM-SHIP2 or SAM-ODIN proteins using the cell spreading assay (Figure 3). The authors should explore the concept that one or both of these molecules are involved in cell spreading, thus dominant-negative forms of these proteins should be tested whether they can block the reduction in cell spreading after ephrinA1 application. This analysis could include also different EphA receptors.

5) Related to this, the authors postulate that specificity of different Ephs for particular downstream SAMs influences their signalling output. Which downstream SAM proteins are recruited in the spreading assay, and does this depend on the receptor type or mutant used? Is SamD5 involved?

6) Data demonstrating that the analysed cells express the different Eph constructs to a similar level is missing. The essay requires staining for Eph, to make sure that the different constructs tested were expressed at a similar level in the cells analysed. It is best to analyse only those cells that express the receptor at a similar level.

---

## [Author Response]

Essential revisions:1) PDB submission codes should be provided.

We have deposited the structure factors and the coordinates of the three structures solved here to the PDB with accession codes of 5ZRX, 5ZRY and 5ZRZ for the EphA2/SHIP2, EphA6/Odin and EphA5/SAMD5 complex structures, respectively. The information has been added in the revised manuscript.

2) A binding constant of 2.2 μM is still rather weak considering cellular concentrations. In principle such weak binding can be functional through multimerization effects (although no multimerization seems to occur with the investigated SAM domains) or higher local concentrations in specific compartments (such as near the membrane). This should be discussed.

Thanks for the great suggestion. It is now well established (thanks largely to wonderful structural studies of the ephrin/Eph ectodomain complex structures) that Eph receptors become multimerized upon ligand engagements. The multimerization of the extracellular domains can bring a number of Eph cytoplasmic tails in close proximity. Additionally, several studies have revealed that SHIP2 can be recruited to plasma membranes via SH2 domain mediated interactions to other cell surface receptors. Both of the above mechanisms raise the local concentrations of Eph and SHIP2, and enhance their binding avidities. We have added this point in the Discussion section in the revised manuscript as follows:

“The binding affinities for all the interactions reported here are in the submicromolar to μM range. […] These two mechanisms can increase the local concentrations of Ephs and their downstream targets, and thus enhance their binding avidities.”

3) Related to this, is there any evidence that SAM domains of different Eph receptors can mediate receptor clustering / heterocomplex formation?

This is a very interesting point. In a paper published in *BBRC* in 2016 (Wang et al., 2016), we characterized each SAM domain of EphB receptors in detail and did not detect any obvious homo- or hetero-dimerization/multimerization of any EphB SAM domains at the assay concentrations up to 200 μM of each SAM domain. See Figure S2 in Wang et al., 2016.

Here we expanded the study to include all the SAMs in the entire Eph receptor family using similar SEC-MALS assay as well as a GST pull down assay. The SEC-MALS results showed that a mixture of the fourteen His-tagged Eph SAM domains do not show any detectable peak shift when it is mixed with another mixture containing thirteen GST-tagged Eph SAM protein mixtures (EphA3 is not included due to severe degradation of GST-EphA3 in our sample preparation). This assay result indicated that no hetero-SAM domain complex can be detected. The SEC-MALS profiles of the His-SAM and GST-SAM mixtures also indicated that no homo-SAM complex could form either (Author response image 1). In an alternative assay, none of the SAM domains in the fourteen His-tagged Eph SAM mixtures could be pulled down by any of the individual thirteen GST-tagged Eph SAM proteins, further showing no hetero-SAM domain interaction could be detected (Author response image 1).

**Author response image 1. respfig1:** Eph receptor SAM domain did not form homo- or hetero-dimers/multimers in solution. (**A**) SEC-MALS analyses showing that mixing the 14 His-tagged Eph receptor SAM domains with the 13 GST-tagged Eph receptor SAM domains (blue line) did not cause detectable elution profile changes when compared to the elution profile of the 14 His-tagged Eph receptor SAM domains (green line) and to that of 13 GST-tagged Eph receptor SAM domains (red line) (i.e. the near perfect overlap of the blue, green and red elution profiles). The concentration of each SAM domain in the assay mixtures is 10 μM. The fitted molar mass values of the His-SAM mixture, GST-SAM mixture, and the His-SAM + GST-SAM mixtures are also indicated in the figure, and further showing no detectable molar mass change upon mixing the two mixtures. (**B**) GST pull-down assay showing that none of the thirteen individual GST-tagged Eph receptor SAM domain had detectable binding to any one of the SAM domain in the 14 His-tagged SAM domain mixture. In this assay, each of purified GST-tagged Eph receptor SAM domain was incubated with a mixtures composed of the 14 His-tagged SAM domains (see 5% input lane). The GSH-Sepharose beads pelleted proteins were separated by Tricine SDS-PAGE and stained with Coomassie blue. No detectable His-tagged SAM domains could be observed in any of the GST-tagged Eph SAM domain pull-down experiments.

4) It would be interesting to study functionally the role of the SAM-SHIP2 or SAM-ODIN proteins using the cell spreading assay (Figure 3). The authors should explore the concept that one or both of these molecules are involved in cell spreading, thus dominant-negative forms of these proteins should be tested whether they can block the reduction in cell spreading after ephrinA1 application. This analysis could include also different EphA receptors.

Thanks for the wonderful suggestion. To address this, we have performed cell spreading assay by transfecting SHIP2-SAM to cells infected with lenti-virus expressing EphA2 or the vector control. The data showed that overexpressing SHIP2-SAM effectively reversed the cell collapse phenotype caused by ephrinA1 activation of EphA2 (compare the third and fourth columns in Figure 3—figure supplement 1), whereas overexpressing SHIP2-SAM did not lead to any observable changes in the GFP-vector expression control (compare the first and the second columns in Figure 3—figure supplement 1), indicating that this blocking effect by SHIP2-SAM is EphA2-dependent. This result is included as a supplementary figure (Figure 3—figure supplement 1) in the revised manuscript.

Regarding different receptors, several other studies have look into this issue. For example, HEK293T cells stably expressing EphA1 spread poorly, and condensed F-actin was seen at the cell periphery by ligand stimulation (Yamazaki et al., 2009). Additionally, treatment of EphA3-transfected HEK293T cells with ephrin-A5 also results in a dramatic contraction of the actin and microtubule cytoskeleton into dense fiber bundles surrounding cell nuclei (Lawrenson et al., 2002). Based on our analysis, both EphA1 and EphA2 have the same interaction partner of SHIP2 and Odin. But for EphA3, we did not find the binding partner yet and the detailed mechanism needs to be figured out in the future. We have also tested EphA5 using the HEK293T cell spreading assay and data showed that in contrast to EphA1 and EphA2, EphA5 did not undergo ephrinA1-induced retraction (Author response image 2). This observation is consistent with the results reported in Figure 3 showing that cells expressing EphA2-SAM_A5_ chimera did not exhibit any EphrinA1 induced retraction. Since this topic has been covered in the earlier literature, we decided not to repeat this in our current manuscript.

**Author response image 2. respfig2:** Representative images showing that HEK293T cell expressing EphA5 WT did not undergo obvious ephrinA1-induced retraction.

5) Related to this, the authors postulate that specificity of different Ephs for particular downstream SAMs influences their signalling output. Which downstream SAM proteins are recruited in the spreading assay, and does this depend on the receptor type or mutant used? Is SamD5 involved?

Thanks for the question. We do not think SAMD5 is involved. First, our biochemical data showed that EphA2 only weakly interacts with SAMD5 (tens of μM; Figure 4D). Second, our cell spreading assay showed that cells expressing EphA2-SAM_A5_ chimera (EphA5-SAM is the specific binding partner for SAMD5) did not exhibit any EphrinA1 induced cell retraction (Figure 3). Third, HEK293T cells expressing EphA5 WT did not undergo ephrinA1 induced retraction (Author response image 2). Based on these observations, we think SAMD5 is unlikely to be involved.

6) Data demonstrating that the analysed cells express the different Eph constructs to a similar level is missing. The essay requires staining for Eph, to make sure that the different constructs tested were expressed at a similar level in the cells analysed. It is best to analyse only those cells that express the receptor at a similar level.

Following the suggestion, we have added a Western blot analysis of different EphA2 proteins alongside each cell-based assay to verify that the expression levels of all EphA2 constructs in the same batch of experiment were similar. See Figure 3C, Figures 3—figure supplement 1C and Figure 3—figure supplement 2C.